# Comprehensive Evaluation of End-Point Free Energy Techniques in Carboxylated-Pillar[6]arene Host–Guest Binding: III. Force-Field Comparison, Three-Trajectory Realization and Further Dielectric Augmentation

**DOI:** 10.3390/molecules28062767

**Published:** 2023-03-19

**Authors:** Xiao Liu, Lei Zheng, Chu Qin, Yalong Cong, John Z. H. Zhang, Zhaoxi Sun

**Affiliations:** 1School of Mathematics, Physics and Statistics, Shanghai University of Engineering Science, Shanghai 201620, China; 2NYU-ECNU Center for Computational Chemistry at NYU Shanghai, Shanghai 200062, China; 3Department of Chemistry, New York University, New York, NY 10003, USA; 4School of Chemistry and Molecular Engineering, East China Normal University, Shanghai 200062, China; 5Shenzhen Institute of Advanced Technology, Chinese Academy of Sciences, Shenzhen 518055, China; 6College of Chemistry and Molecular Engineering, Peking University, Beijing 100871, China

**Keywords:** pillar[*n*]arenes, host–guest binding, force field, dielectric constant, three-trajectory realization

## Abstract

Host–guest binding, despite the relatively simple structural and chemical features of individual components, still poses a challenge in computational modelling. The extreme underperformance of standard end-point methods in host–guest binding makes them practically useless. In the current work, we explore a potentially promising modification of the three-trajectory realization. The alteration couples the binding-induced structural reorganization into free energy estimation and suffers from dramatic fluctuations in internal energies in protein–ligand situations. Fortunately, the relatively small size of host–guest systems minimizes the magnitude of internal fluctuations and makes the three-trajectory realization practically suitable. Due to the incorporation of intra-molecular interactions in free energy estimation, a strong dependence on the force field parameters could be incurred. Thus, a term-specific investigation of transferable GAFF derivatives is presented, and noticeable differences in many aspects are identified between commonly applied GAFF and GAFF2. These force-field differences lead to different dynamic behaviors of the macrocyclic host, which ultimately would influence the end-point sampling and binding thermodynamics. Therefore, the three-trajectory end-point free energy calculations are performed with both GAFF versions. Additionally, due to the noticeable differences between host dynamics under GAFF and GAFF2, we add additional benchmarks of the single-trajectory end-point calculations. When only the ranks of binding affinities are pursued, the three-trajectory realization performs very well, comparable to and even better than the regressed PBSA_E scoring function and the dielectric constant-variable regime. With the GAFF parameter set, the TIP3P water in explicit solvent sampling and either PB or GB implicit solvent model in free energy estimation, the predictive power of the three-trajectory realization in ranking calculations surpasses all existing end-point methods on this dataset. We further combine the three-trajectory realization with another promising modified end-point regime of varying the interior dielectric constant. The combined regime does not incur sizable improvements for ranks and deviations from experiment exhibit non-monotonic variations.

## 1. Introduction

End-point free energy techniques as pivotal tools in drug discovery are commonly recognized as regimes with intermediate accuracy and efficiency lying between molecular docking and (alchemical) free energy calculation. In protein–ligand and protein–protein interactions, such a computational ladder is believed to be solid, with a huge library of case reports supporting its usage [1,2,3,4,5,6,7,8]. However, such ‘common sense’ seems broken in host–guest complexes, although the latter are often recognized as prototypical models of biomacromolecular systems. Specifically, although rigorous free energy simulations still achieve better performance than end-point methods [9,10,11], docking calculations could produce rather small deviations from experimental reference compared with other costly free energy techniques [12]. The deviations of end-point calculations are unacceptably huge, making them practically useless for host–guest binding. As a testing bed of common interest, the carboxylated-pillar[6]arene (WP6) host–guest dataset in the recent SAMPL9 challenge has been employed to benchmark the performance of various end-point procedures in our recent works, where extensive numerical data suggest the unsuitability of standard end-point calculation for host–guest binding [13,14]. To provide really meaningful predictions, alterations seem necessary. In the context of protein–ligand binding, there are a spectrum of modifications of the end-point procedure in order to achieve higher accuracy. However, whether these altered regimes are practically useful in host–guest binding requires detailed evaluations, as experiences accumulated in protein–ligand complexes (e.g., the accuracy ladder of computational techniques) have been proven non-transferable to host–guest systems. One altered regime that achieves pronounced improvements for both absolute and relative values is regression, and the dielectric constant-variable regime that is widely employed in end-point free energy calculations performs rather poorly and marginally betters the ranking calculation [13]. The underperformance of the dielectric-constant regime is to some extent conflicting with protein–ligand situations, which again hints at the different behaviors of host–guest and biomacromolecular systems. 

There are generally two sampling protocols in end-point free energy calculations. The first one, the ‘normal’ and popular single-trajectory realization, samples only the protein-ligand/host–guest bound state and extracts free energy of binding by computing inter-molecular potentials, solvation and entropic contributions. The main motivation of this single-trajectory realism is the exact cancellation of internal energies, and the end-point sampling only needs to converge the fluctuations in inter-molecular components. In complexes involving biomacromolecular species, the internal energies summed over a large amount of atoms are huge in magnitude, and the fluctuations in internal energies are overwhelmingly high. However, the smaller sizes of host–guest systems lead to smaller internal energies and fluctuations, which makes them potentially suitable for the second end-point sampling protocol named the three-trajectory realization. As suggested by its name, the three-trajectory realization samples three ensembles including both the bound and unbound states [15,16,17,18], with the latter containing two components of the protein/host-only and ligand/guest-only systems. The three-trajectory realization computes free energy of binding as the difference between the bound-state energetics and the unbound-state results, which requires the evaluation of all potential terms and thus the convergence of intra-molecular energetics also. As the intra-molecular energetics are strongly dependent on force-field parameters, some detailed understandings of the force-field behaviors are pivotal to guide practical use. Therefore, the commonly applied force-field selections of general AMBER force field (GAFF) derivatives [19] are investigated comprehensively in terms of force-field definitions and dynamic behaviors. Differences between GAFF and GAFF2 for the macrocyclic host are identified in many aspects, which could have significant impacts on the end-point estimates of binding affinities. As we only focus on the charge scheme, water model, and docking procedure in model construction and overlook this critical GAFF-version comparison in our previous works [13,14], we add a benchmark for the popular GAFF versions in single-trajectory end-point calculations as a crucial addition to the influencing factors in standard end-point calculations. Then, the three-trajectory realization of end-point free energy calculations under transferable GAFF derivatives is conducted and a detailed investigation of energetic behaviors is presented. Numerical results suggest that the three-trajectory realization serves as a promising modification for ranking predictions. Finally, the three-trajectory realization is combined with the dielectric constant-variable regime, producing a more advanced modification. The numerical behavior of this newly constructed dielectric-constant-augmented three-trajectory regime is benchmarked at physiochemically reasonable dielectric constants. 

## 2. Results and Discussions

### 2.1. A Detailed View of GAFF Derivatives

The single-trajectory realization of end-point free energy calculations only samples the host–guest complex in the bound state, where the host cavity is occupied by the guest molecule due to the formation of favorable host–guest coordination. Thus, it is unlikely to observe a squashed host or a significant conformational change, and the intra-molecular interactions (especially bonded parameters) could be of relatively little relevance. By contrast, the three-trajectory realization requires the sampling of both the bound complex and the free/unbound host-only and guest-only systems. In the unbound state (free host and free guest in solvent), the host–guest coordination is replaced by host/guest–water and also host/guest–ion interactions. If force field parameters do not produce a balanced description of the (de)solvation phenomenon and the backbone stiffness, the water- and ion-filled host cavity could possibly be squashed in the unbound state. Similar observations have been reported in our previous works on other host molecules including cucurbiturils and cyclodextrins [20,21,22]. This phenomenon indicates the existence of a conformational change in the host upon the replacement of solvents by the guest, which is often accompanied by dramatic drifts and fluctuations in system energetics (especially internal energies). Considering this fact, the three-trajectory realization should be more sensitive to intra-molecular interactions than the popular single-trajectory regime. Therefore, we perform an extensive evaluation of the commonly used transferable parameters of GAFF derivatives in the current WP6 host–guest dataset. 

Two GAFF derivatives are commonly employed in modern computational investigations. The earlier GAFF v1.81 (or simply GAFF) is a parameter set and minorly updated from the original GAFF, while a newer GAFF version 2.11, or simply GAFF2, is the latest component of the second generation of GAFF, with majorly re-fitted transferable parameters. We first present a detailed comparison between the two GAFF versions in order to identify their differences and similarities. The main components of intra-molecular interactions, the bonded terms, include bond stretching, angle bending, and torsional potentials. Among them, the torsional terms contribute the largest portion of variations during conformational rearrangements in the presence of bond-length constraints. Through a detailed comparison between definitions of torsional terms in GAFF and GAFF2, we identify two terms that exhibit noticeable differences, as shown in Figure 1a,b. By contrast, all terms presented in Appendix A share exactly the same parameters under the two GAFF derivatives. The torsional term in Figure 1a has a barrier height ~0.9 kcal/mol in GAFF, while in GAFF2 the barrier height is elevated by a factor of ~2. The term in Figure 1b does not have an explicit definition in GAFF (i.e., no torsional barrier) but is represented by three Fourier-series terms with different periodicities (one, two, and three), phases, and barrier heights in GAFF2. As the two differing terms are not relevant to the stiffness of the host ring but are only determining the rotation of -CH_2_-COO^−^ tails, the stiffnesses of the host cavity produced by the two GAFF derivatives are expected to be similar, but the orientations of the polar -CH_2_-COO^−^ tails are expected to exhibit different behaviors. We have compared the other bonded terms (i.e., bond-stretching and angle-bending potentials) of GAFF derivatives and also identified noticeable differences. For instance, the averaged variation of the bond-stretching force constants is ~15%, but the equilibrium length is minimally perturbed (~0.02% on average). Aside from the bonded parameters, non-bonded terms are also entering intra-molecular interactions between atoms separated by more than two chemical bonds. The 1–4 scaling constants in the two GAFF versions are the same, but the other parameters do show some differences. As the atomic charges are obtained with molecule-specific fitting, in the constructed models only the vdW terms differ. Through a detailed comparison of vdW terms, we confirm that all vdW parameters (i.e., both the size parameter σ and the energy parameter ε) are altered in GAFF2 compared with GAFF, but the magnitudes of the variations are insignificant. For example, the difference between the GAFF and GAFF2 σ averaged over all host atoms is ~2%. As these differing terms of bond-stretching, angle-bending, and intra-molecular vdW interactions are not expected to have significant influences on conformational preferences or be involved in substantial reorganizations/rearrangements upon host–guest binding, the main difference between GAFF and GAFF2 would be triggered by the torsional terms in Figure 1a,b. 

### 2.2. Dynamic Behaviors of the Macrocyclic Host

The detailed analyses of the force field parameters and energetics presented above are informative, providing hints on the similarities and differences between different parameter sets. However, whether these factors are directly impacting the dynamics of the macrocyclic host remains to be revealed. Thus, we perform unbiased sampling in explicit solvent in order to probe the dynamic behaviors produced by both parameter sets. Note that the counter ions (sodium cations) are added for neutralization. Thus, the host cavity could be occupied by water molecules and/or cations, forming host–water or host–ion coordination. The solvated host experiences 5000 steps minimization, 0.1 ns NVT equilibration, and then 1 ns NPT equilibration. After that, the 500 ns NPT production run with a sampling interval of 0.5 ns is spawned. 

The superpositions of unbiased trajectories are presented in Figure 1c,d. Concerning the stiffness of the host backbone, we do not observe noticeable differences between GAFF and GAFF2, which is consistent with the observation in the previous term-specific comparison. However, the rotational dynamics of the -CH_2_-COO^−^ rims do exhibit significant differences. Under GAFF, the WP6 host does not exhibit any conformational preference and its -CH_2_-COO^−^ tails rotate freely. By contrast, a directional rotational preference featuring a fraction of -CH_2_-COO^−^ tails stabilizing at the cavity entrance could be identified from the GAFF2 overlay. This difference in the dynamic behaviors of -CH_2_-COO^−^ tails agrees with the detailed analyses of torsional terms in the previous section, where the GAFF2 parameter set produces a more regular description of the -CH_2_-COO^−^ rotation than GAFF. By including the solvent and ions in the visualization, we observe that this unusual rotational behavior is related to the formation of host–cation coordination. Thus, the regular/localized behavior under GAFF2 is caused by the interplay of host–solvent/ion interactions and intra-molecular torsional potentials. The host–solvent/ion coordination also exists under GAFF but does not incur significant conformational change like the GAFF2 situation. Overall, the dynamic behaviors of the WP6 host have been confirmed to be different under the two popular GAFF derivatives, which would obviously impact the sampling trajectories in end-point free energy calculations.

### 2.3. GAFF vs. GAFF2 in Single-Trajectory Estimates

Crucial influencing factors of model construction in end-point calculations investigated in our previous works include the charge scheme, water model, and docking procedure for the initial condition, all of which do not have significant impacts on the performance of end-point calculations given a selected implicit solvent model [13,14]. However, in the current work we identify another critical influencing factor—the version of the transferable GAFF derivatives. The energetics and the dynamic behavior of the macrocyclic host WP6 exhibit noticeable differences, which makes the selection of transferable parameter sets for bonded and vdW parameters also an influential factor. Therefore, here we perform single-trajectory end-point free energy calculations with the host described by GAFF and GAFF2 as an additional benchmark supplementing to our previous paper on the ‘standard’ procedure [14]. As the GAFF-host GAFF2-guest model is newly constructed in the current work and new simulations last 300 ns, which is much longer than the 100-ns length in our previous work, in the current additional single-trajectory benchmark we also employ the 300-ns sampling length for the GAFF2-host GAFF2-guest model. The number of snapshots for the NMA calculations in the previous work (50 structures) is also doubled here (i.e., 100) to be consistent with the three-trajectory protocol. The numerical results of the new single-trajectory estimates are summarized in Table 1 (MM/PBSA) and Table 2 (MM/GBSA), along with quality metrics of the root-mean-squared error (RMSE), mean signed error (MSE), Kendall τ rank coefficient [23], and Pearlman’s predictive index (PI) [24]. 

The experiment–calculation correlation under GAFF and GAFF2 is presented in Figure 2a,b. The 100 ns GAFF2 results reported in our previous work [14] are also given in the inset of each subplot for comparison. For most systems, there is no significant difference between the previous 100 ns and the new 300 ns GAFF2 results. Only a small fraction of systems exhibits noticeable differences, but the rank of different systems is not obviously varied. By comparing the quality metrics (both error estimate and ranking coefficient) in Table 1 and Table 2 with our previous work [14], no significant difference could be observed. Therefore, the calculation accuracy is not changed when shifting from the previous 100-ns trajectory to the lengthened 300-ns one, validating the convergence of the free energy calculation. As for the comparison between the 300 ns GAFF and GAFF2 estimates, we identify obvious differences and thus conclude that the confirmed difference in the host parameters would incur noticeable differences in the estimated end-point binding affinities. 

We then turn to the cross-comparison of quality metrics in Figure 2c,d to obtain a statistical evaluation of the performance. Still, we first check the consistency of the performance statistics of the current 300 ns results and the previous 100 ns ones under GAFF2. For water models, the TIP3P solvation systematically worsens the prediction quality compared with SPC/E, and the PB implicit solvent treatment behaves less satisfactorily than GB. Both trends are the same as the previous report [14], validating the reliability of our calculation. As for the relative performance of GAFF and GAFF2, the GAFF2 parameter set prevails for both the reproduction of absolute values (RMSE in Figure 2c) and the ranking information (τ in Figure 2d), regardless of the explicit-water model in sampling and the implicit solvent model in free energy estimation. Thus, in single-trajectory end-point free energy calculations with the popular ‘standard’ procedure, the GAFF2 parameter set could be recommended for the pillararene host.

### 2.4. Three-Trajectory End-Point Free Energy Estimates

The three-trajectory realization samples both the bound and unbound states. The detailed energetic contributions in individual ensembles are presented in Appendix A, and the final estimates of binding affinities are given in Table 3 and Table 4. The first observation of the final estimates concerns the magnitude of the fluctuations. By comparing the single-trajectory estimates in Table 1 and Table 2 and the three-trajectory results in Table 3 and Table 4, it can be easily identified that the binding free energies provided by the latter regime are noisier (larger uncertainties), which is an expected behavior according to the existing experience in protein–ligand binding due to the inclusion of internal energies in free energy estimation. However, an interesting observation unexpected here is that the fluctuation magnitude of the solvated host is much larger than the other (i.e., bound host–guest complex and unbound solvated guest), as shown in the detailed state-specific components in Appendix A. This observation suggests that the unbound host is more difficult to be sampled, and thus should be distributed much more sampling times with given computing resources. To further elucidate the large fluctuations in the solvated host ensemble, we compare individual energy terms and identify the polar contribution (i.e., the implicit solvent and electrostatic terms) as the main reason. As illustrative examples, we select the GAFF2 parameter set for the WP6 host and TIP3P for water and monitor the bound and unbound ensembles involving guests G01, G06, and G12. The time series data of PB/GB terms for the host–guest and solvated guest systems are shown in Appendix A, while those of the total polar contribution (i.e., the sum of the implicit solvent and electrostatic terms) are given in Appendix A. For all of the three guests, the implicit solvent terms and the total polar contribution (and thus also the electrostatics) in both host–guest and solvated guest ensembles exhibit equilibrium fluctuating behaviors, leading to their small uncertainties in free energy estimation. However, the PB/GB and electrostatics time series for the solvated host in Appendix A exhibit huge fluctuations and systematic drifts. The variation in the GB term almost compensates that of the intra-molecular electrostatics, leading to the smaller fluctuation in the total polar contribution shown in Appendix A. However, the magnitude of the PB variation does not match with electrostatics, leading to the huge fluctuation in the PB+ELE curve in Appendix A. For this total polar contribution, such a range from −800 kcal/mol to ~0 kcal/mol is unexpectedly huge, and the systematic variation suggests that the sampling in the host-only state should be non-convergent. Considering the length of our sampling (300 ns), it seems practically impossible to converge this implicit solvent contribution with modern standards of end-point sampling (several ns to 1 μs). To further investigate this unusual behavior, we extract structures at representative points. It is observed that the host molecule is experiencing significant intra-molecular conformational fluctuations, and in many regions/snapshots the -CH_2_COO^−^ tails stay at the entrance of the host entrance. As discussed in the comparison of host dynamics in Section 3.2, these structures are related to the formation of host–ion coordination and also the regular torsional potentials under GAFF2. We further depict the solvent-excluded surface probed using the solvent probe in Appendix A. The initial condition has a fully opened cavity with the interior accessible to solvents. As the sampling proceeds, the host cavity is rearranged to be partially and finally fully closed due to the rotation of the substituting -CH_2_COO^−^ groups. This structural rearrangement of the host surface definitions serves as a critical factor causing the huge variations in the implicit solvent contributions (and thus also the intra-molecular electrostatics). In Appendix A, we present the implicit solvent terms in GAFF sampling and observe fluctuations less significant than the GAFF2 case. Thus, the unusual fluctuating behavior in the host-only simulation is related to the GAFF2 parameter set. It should be noted that such a huge force-field dependence is generally not observed in protein–ligand complexes, which again validates our conclusion on the dissimilarities of host–guest and protein–ligand systems and emphasizes that the existing knowledge accumulated in protein–ligand binding cannot be directly transferred to host–guest systems. 

We then turn to the comparison of the calculated binding affinities. The correlograms of the three-trajectory estimates under different parameter sets are given in Figure 3a,b. With a given implicit solvent model, the estimates under different parameter sets differ significantly in magnitude, but the relative sizes of different host–guest pairs seem similar. This systematic over-/under-estimation of binding affinities (depending on the sign) is also suggested by the MSE shown in Table 3 and Table 4. Such an offset phenomenon is especially related to the implicit solvent contribution in the host-only simulations. Considering the huge differences between the three-trajectory estimates and the experimental references, comparing the absolute values of the binding affinities could not be meaningful (c.f., RMSE in Figure 3c). Therefore, the useful predictions with the three-trajectory realization could mainly be the ranking information. In Figure 3d, the τ values of the three-trajectory realization are ~0.4, which are comparable to existing predictions given by PBSA_E scoring (either pre-fitted with protein–ligand data or refitted with WP6-specific information) and the dielectric constant-variable regime. The best performances are achieved under the GAFF parameter set, the TIP3P water in sampling, and PB in free energy estimation, and the highest ranking τ ~0.51 surpasses all existing end-point reports and even many alchemical calculations [13,14]. Therefore, the three-trajectory realization indeed serves as a promising modification of the end-point procedure. A phenomenon worth noting is that GAFF performs better than GAFF2 in the current three-trajectory realization, while the opposite is observed in the popular single-trajectory regime (see the last paragraph of the previous section). The relative performance of different force fields should be related to error cancellation. 

### 2.5. Dielectric-Constant-Augmented Three-Trajectory Estimates

Considering the successes in predicting the ranking information with both the three-trajectory realization and the dielectric constant-variable regime in the single-trajectory case, we combine the two non-conflicting regimes to form a more advanced modification, i.e., the three-trajectory dielectric constant scheme. We select three parameter combinations to test this method, including the best-performing set in the three-trajectory realization GAFF+TIP3P (see Figure 3d), the GAFF+SPC/E set to check the variation of the water model, and the GAFF2+SPC/E set that performs best in the single-trajectory realization (see Figure 2c,d and our previous work [14]). The interior dielectric constants tested include physiochemically reasonable values of 1, 2, 4, and 6.

The affinity variations of Individual host–guest pairs under two of the three parameter sets (GAFF+SPC/E and GAFF2+SPC/E) are presented as illustrative examples in Figure 4a,b. Interestingly, with the increase in the interior dielectric constant, the binding affinities of all host–guest pairs increase monotonically. This phenomenon is in stark contrast to the single-trajectory case reported in our previous work [13], where system-dependent responding behaviors are observed. To obtain a deeper understanding about the response to dielectric constant variations, we do some contribution analysis here. The dielectric constant-dependent components in the single-trajectory estimates are inter-molecular electrostatic and implicit solvent contributions,
(1)$$E_{1-\mathrm{traj}}\left(\varepsilon_{\mathrm{in}}\right)={\langle E_{\mathrm{ele},\;\mathrm{inter},\;\mathrm{host}-\mathrm{guest}}\rangle }_{\mathrm{bound}}+{\langle E_{\mathrm{GB},\;\mathrm{host}-\mathrm{guest}}-E_{\mathrm{GB},\;\mathrm{host}}-E_{\mathrm{GB},\;\mathrm{guest}}\rangle }_{\mathrm{bound}}$$
while the situation is slightly complicated in the three-trajectory case. In the bound state, both the intra- and inter-molecular contributions ($E_{\mathrm{bound}}\left(\varepsilon_{\mathrm{in}}\right)={\langle E_{\mathrm{ele},\;\mathrm{inter},\;\mathrm{host}-\mathrm{guest}}+E_{\mathrm{ele},\;\mathrm{intra},\;\mathrm{host}-\mathrm{guest}}\rangle }_{\mathrm{bound}}+{\langle E_{\mathrm{GB},\;\mathrm{host}-\mathrm{guest}}\rangle }_{\mathrm{bound}}$) are involved, while in the unbound state only intra-molecular terms ($E_{\mathrm{unbound}}\left(\varepsilon_{\mathrm{in}}\right)={\langle E_{\mathrm{ele},\;\mathrm{intra},\;\mathrm{host}}+E_{\mathrm{GB},\;\mathrm{host}}\rangle }_{\mathrm{unbound}}+{\langle E_{\mathrm{ele},\;\mathrm{intra},\;\mathrm{guest}}+E_{\mathrm{GB},\;\mathrm{guest}}\rangle }_{\mathrm{unbound}}$) exist. The dielectric constant-dependent terms in the three-trajectory estimate can thus be written as
(2)$$\begin{array}{l} E_{3-\mathrm{traj}}\left(\varepsilon_{\mathrm{in}}\right)=E_{\mathrm{bound}}\left(\varepsilon_{\mathrm{in}}\right)-E_{\mathrm{unbound}}\left(\varepsilon_{\mathrm{in}}\right)\\ ={\langle E_{\mathrm{ele},\;\mathrm{inter},\;\mathrm{host}-\mathrm{guest}}+E_{\mathrm{ele},\;\mathrm{intra},\;\mathrm{host}-\mathrm{guest}}\rangle }_{\mathrm{bound}}+{\langle E_{\mathrm{GB},\;\mathrm{host}-\mathrm{guest}}\rangle }_{\mathrm{bound}}\\ -{\langle E_{\mathrm{ele},\;\mathrm{intra},\;\mathrm{host}}+E_{\mathrm{GB},\;\mathrm{host}}\rangle }_{\mathrm{unbound}}-{\langle E_{\mathrm{ele},\;\mathrm{intra},\;\mathrm{guest}}+E_{\mathrm{GB},\;\mathrm{guest}}\rangle }_{\mathrm{unbound}} \end{array}$$
which can be rearranged as
(3)$$\begin{array}{l} E_{3-\mathrm{traj}}\left(\varepsilon_{\mathrm{in}}\right)={\langle E_{\mathrm{ele},\;\mathrm{inter},\;\mathrm{host}-\mathrm{guest}}\rangle }_{\mathrm{bound}}\\ +{\langle E_{\mathrm{ele},\;\mathrm{intra},\;\mathrm{host}-\mathrm{guest}}\rangle }_{\mathrm{bound}}-{\langle E_{\mathrm{ele},\;\mathrm{intra},\;\mathrm{host}}+E_{\mathrm{ele},\;\mathrm{intra},\;\mathrm{guest}}\rangle }_{\mathrm{unbound}}\\ +{\langle E_{\mathrm{GB},\;\mathrm{host}-\mathrm{guest}}\rangle }_{\mathrm{bound}}-{\langle E_{\mathrm{GB},\;\mathrm{host}}+E_{\mathrm{GB},\;\mathrm{guest}}\rangle }_{\mathrm{unbound}} \end{array}$$

The first term in Equation (3) denotes the three-trajectory bound-state inter-molecular electrostatic contribution, which is exactly the same as that in Equation (1). The second row includes the variation in intra-molecular interactions upon the binding event, and is not zero (cancelled out) due to the incorporation of the unbound state in the free energy calculation. This is also the term that the three-trajectory realization is believed to differ significantly from in the single-trajectory form. The third row, the implicit solvent contribution upon the binding event, corresponds to the second term in Equation (1) (substituting the subscript unbound with bound). Compared with the single-trajectory case, the incorporation of the unbound-state sampling alters this implicit solvent contribution to some extent. Comparing Equation (1) with Equation (3), we know that the differences between the single-trajectory and three-trajectory responses to variations in the interior dielectric constant are caused by the intra-molecular electrostatics and the implicit solvent contribution. Then, an interesting question to ask is which contribution (i.e., electrostatic or implicit solvent) is playing a more critical role in the three-trajectory realization. To investigate this issue, we extract the detailed components in both the bound and unbound states and observe that the negative implicit solvent term is larger in magnitude than the electrostatic counterpart for all systems investigated (i.e., $\langle E_{\mathrm{ele},\;\mathrm{intra}}+E_{\mathrm{GB}}\rangle <0$ see PB/GB+ELE curves in Appendix A), and the same applies to their responses to the variation in the dielectric constant (expected considering the analytical derivative of, e.g., charge–charge electrostatics). Further, the difference presented in the third row of Equation (3) is larger in magnitude than the second row. As a result, with the increase in the internal dielectric constant, their sum increases monotonically and makes the net binding affinity more positive. Overall, in the three-trajectory end-point calculation that incorporates the conformational change upon binding/unbinding, the implicit solvent term plays a more crucial role during electrostatics screening/scaling with the dielectric constant-variable regime. The resulting responding behavior thus differs from the single-trajectory situation.

We then turn to the quality metrics to evaluate whether the coupling of dielectric constant variations with the three-trajectory sampling could better the performance. The dielectric constant-dependent RMSE and τ curves are presented in Figure 4c,d. Concerning the reproduction of the absolute values of binding affinities, the RMSE of the GAFF2+SPC/E parameter combination decreases monotonically along the dielectric constant variation, which is caused by the monotonically increasing trend of the absolute values (c.f., Figure 4a,b) and the systematic overestimation of the binding strength under this parameter set (see Figure 3b). Namely, the free energy variations in response to the internal dielectric constant are compensating the offset in the original binding free energy estimates. Unlike the GAFF2+SPC/E case, the GAFF+SPC/E parameter set exhibits a non-monotonic response, which can be understood in a similar way. Namely, the binding affinities under this parameter combination are first increased and become close to the experimental reference, causing a decrease in RMSE. Then, the binding affinities are increased further and depart from the experimental range again, which again elevates the error size. The other parameter combination, GAFF+TIP3P, exhibits a monotonic increasing behavior, differing from the other two examples. Such parameter-dependent responses of the three-trajectory curves somehow differ from the single-trajectory one. For example, comparing the single- and three-trajectory curves under the same GAFF2+SPC/E parameter combination, the three-trajectory one exhibits a monotonic decreasing behavior, while the single-trajectory one increases monotonically. Thus, experiences accumulated in the single-trajectory end-point calculations cannot be directly extended to the three-trajectory situation, due to the different physics therein. As for the calculation of the ranking information, the adjustment of the interior dielectric constant does not significantly perturb the quality of the three-trajectory estimates. The maximum change in τ for the three examples is ~0.05. By contrast, the single-trajectory calculation experiences dramatic performance improvements in the scanned range. The different responses to the adjustment of the dielectric constants for ranking information again emphasize that the experiences accumulated in the popular single-trajectory realization cannot be generalized to the three-trajectory case. Another interesting observation about the ranking–coefficient curve is the location of the maxima. For all of the three parameter combinations, τ reaches its maximum at dielectric constant 2. The maximum value of Kendall τ for the best-performing GAFF+TIP3P parameter set is 0.54, which surpasses all existing end-point predictions on this dataset and is comparable to the costly alchemical method reported by others [13,14]. Rank predictions of such qualities already seem acceptable in modern computational biophysics, suggesting that the three-trajectory regime and its dielectric constant alteration are promising regimes for host–guest binding. 

## 3. Computational Details

### 3.1. Model Construction

The model construction follows our WP6 end-point series works [13,14]. The 3D chemical structures of the host and guest molecules shown in Figure 5 are obtained from the GitHub site of the SAMPL9 challenge [25]. Due to the observed similar prediction qualities of commonly employed charge schemes in this dataset [13,14], we generate atomic charges with the restrained electrostatic potential (RESP) [26] scheme with ESP data scanned at the traditional HF [27,28,29] /6-31G* level (i.e., RESP-1 charge set in our previous works and our GitHub repository for atomic charges https://github.com/proszxppp/WP6-host-guest-binding, accessed on 22 February 2023). As for the other parameters (e.g., bonded terms), following our previous works [13,14], we take the transferable GAFF2 parameter set [19] and a recent publication extending the GAFF2 parametrization to Si-involved species [30], forming a GAFF2-host GAFF2-guest modelling protocol. According to the detailed comparison of GAFF derivatives shown later in the results section, there are noticeable differences between GAFF and GAFF2 for the macrocyclic host WP6, which would influence the end-point outcome, especially for the three-trajectory realization considered here. Therefore, we also construct a GAFF-described host and combine it with GAFF2-described guest molecules in the current modelling, forming another modelling protocol (GAFF-host GAFF2-guest). Solvation with a 12 Å solute-edge distance is performed with two popular three-point models of TIP3P [31,32] and SPC/E [33], which differ in the parameter space and produce different bulk and solvation behaviors. For host–guest complexes, the simulation needs a starting point or initial condition, which is prepared by molecular docking with Autodock Vina [34]. Specifically, the top-1 bound structure obtained with the Autodock4 scoring function is selected [34,35]. Non-polarizable monovalent spherical counter ions [36,37] of Na^+^ or Cl^−^ are added to neutralize the simulation cell. Periodic boundary conditions are employed to replicate the unit cell in the whole space. In the modelling of host–guest complexes, the combination of the GAFF version and the solvent model gives four protocols for model construction. 

Aside from the solvated host–guest complex, the unbound state containing the decoupled host–guest pair is also sampled in the three-trajectory realization of end-point free energy calculation. This leads to two independent simulations, each of which contains either the solvated host or the guest. The modelling protocol of these unbound systems is similar to the complex system. Namely, we use the HF/6-31G*-targeted RESP charges to describe electrostatics, employ GAFF or GAFF2 for host and GAFF2 for guest, and add solvents with either of the two water models and monovalent counter ions for neutralization.

### 3.2. Sampling and Free Energy Estimation

The constructed solvated host, guest, and host–guest complex first experience 5000 steps minimization, 300 ps constant-volume heating with weak harmonic restraints on heavy atoms of solutes, and 1-ns NPT relaxation to reach an equilibrated state at 300 K and 1 atm. The sampling time of host–guest complexes in our previous works is 100 ns, which is already long in end-point free energy calculations. However, the fluctuations in the free energy estimates from the three-trajectory realization are often much larger than the single-trajectory counterpart due to the inclusion of the reorganization free energy (or equivalently the internal energy) [15,16,38], triggering more difficulties in converging the host–guest binding affinities. Thus, we triple the sampling length for host–guest complexes (i.e., 300 ns). For the host-only and guest-only simulations, this sampling length is also applied. The sampling interval remains the same as our previous work, i.e., 10 ps. Bonds involving hydrogen are constrained with the SHAKE algorithm [39,40], and the time step of 2 fs is employed. We use Langevin dynamics [41] with the collision frequency of 2 ps^−1^ for temperature regulation, and the Berendsen barostat with isotropic scaling for pressure regulation. The cutoff for non-covalent interactions is set to 8 Å (AMBER default), and the PME method [42] is used to treat long-range electrostatics. The hybrid-precision GPU engine of AMBER[43] is used for dynamics propagation. 

The three-trajectory end-point method estimates the free energy of binding by taking the difference between the bound and unbound states. The former corresponds to the solvated host–guest complex, while the latter is sampled with the host-only and the guest-only simulations. The calculation of the energetics of the individual systems involves the gas-phase force-field energy estimation, the implicit solvent treatment, and normal mode analysis (NMA) [44] for the entropic contribution. All extracted snapshots (30,000) are incorporated in gas-phase and implicit solvent calculations. The implicit solvent contributions include the polar part estimated with either PB [45] or the popular GB^OBC^ with the second set of modified Bondi radii [46,47] and the non-polar solvent-accessible surface area contribution [48]. As for the NMA step, due to the high cost and the relatively insignificant conformation-dependence, we only include 100 equally spaced snapshots in calculations, i.e., 3-ns intervals. 

## 4. Concluding Remarks

Host–guest binding remains rather challenging in the modern computational community. End-point free energy calculations with established predictive power in protein–ligand complexes are not well understood in host–guest modelling. In our series works towards a thorough evaluation of host–guest modelling focusing on the latest SAMPL9 host–guest dataset involving a pillararene derivative, contrary to the over-optimistic view reported in numerous reports published in mainstream forums, we prove with solid numerical evidence that standard end-point free energy calculations are practically useless and achieve prediction qualities far from acceptable. Modifications are necessary to bring the computational results close to the experimental reference. The alterations considering linear regression and interior dielectric constant have been extensively validated in our previous works. In this paper, we focus on the specific modification of the three-trajectory realization. 

The three-trajectory realization is rather uncommon due to the overwhelming fluctuations in internal energies in protein–ligand complexes but could be well-suitable for host–guest binding due to the relatively small sizes of the components involved in inter-molecular packing and thus the small energy fluctuations. Force field parameters, especially bonded terms, play a critical role here, and a thorough investigation of the similarities and differences between the popular GAFF derivatives is presented. The GAFF derivatives are found to share similar parameters for the pillararene backbone, and the differences mainly lie in torsional terms involving -CH_2_COO^−^ tails. Directly applying the transferable parameter sets in dynamics simulation, we observe similar backbone stiffnesses for WP6 under GAFF and GAFF2, but the polar tails at both rims exhibit different conformational preferences. Structural analysis reveals that the directional orientation in the solvated host is related to ion coordination (i.e., host–ion binding). Overall, the observed different dynamic behaviors produced by the two GAFF derivatives are expected to produce different conformational preferences in both the unbound host and the bound host–guest complex, and thus produce different fluctuating behaviors of energetics. 

Due to the observed significant differences between GAFF derivatives, this force-field selection could be a crucial influencing factor determining the prediction quality of host–guest binding affinity. As this conclusion could be applicable to both single-trajectory and three-trajectory realizations, we thus first benchmark single-trajectory end-point calculations under GAFF and GAFF2 as a critical addition to the extensive benchmark calculations presented in our previous work. The numerical results support the usage of the SPC/E water model, the GB implicit solvent, and the GAFF2 parameter set. Thus, in the popular single-trajectory realization, GAFF2 outperforms GAFF for this host family. 

The three-trajectory end-point free energy calculations exhibit behaviors somehow different from the single-trajectory procedure. First, the shift to the three-trajectory realization further triggers an increase in energetic fluctuations and noisier statistics, which calls for a lengthened sampling time in order to minimize the statistical uncertainty to values similar to the single-trajectory estimates. A detailed look into the numerical data suggests that the uncertainty/fluctuation increase is mainly caused by the inclusion of the unbound states, especially the host-only system. Based on this fact, a general rule or recommendation for the three-trajectory end-point free energy calculation is to perform longer sampling in the unbound host-only state, while the unbound guest and the bound host–guest states could be less focused. Second, the comparison between the two GAFF versions suggests that the energetic fluctuations under GAFF2 are much more significant than GAFF. This difference in fluctuating behaviors could be specifically located on the implicit solvent energetics of the unbound host, which is highly related to the directional rotation of the -CH_2_COO^−^ rims. Third, the predicted binding affinities under the three-trajectory realization exhibit systematic overestimation of the binding strength also due to the inclusion of the unbound-state sampling (more specifically the implicit solvent contribution). This offset leads to significantly different RMSE values for different parameter sets but does not poison the ranking calculation. The best-performing parameter combination GAFF+TIP3P achieves a Kendall τ ~0.5, which surpasses all end-point predictions on this dataset and is even comparable to the costlier alchemical method. 

To pursue a further improved performance and accumulate experiences of the three-trajectory realization, we combine two non-conflicting modified end-point procedures into a three-trajectory dielectric constant protocol. This newly constructed modification does not systematically better or worsen the reproduction of the absolute binding affinities, which differs from the single-trajectory situation. As for the ranking calculation, the combined regime leads to minor but not systematic improvements compared with individual modifications (i.e., either the three-trajectory realization or the single-trajectory dielectric constant protocol). Further, the prediction quality of the ranking information achieves the best performance when the interior dielectric constant reaches 2, which also differs from the single-trajectory case at ~6 dielectric constants. Overall, combining different non-conflicting protocols into more advanced end-point schemes could be helpful, but the improvements over individual baseline alterations could be marginal. To further explore the space of advanced end-point techniques and search for really usable protocols, we would attempt a series of other modifications (e.g., enhanced sampling [10,20,21,22,49] and the multi-scale treatment [50,51]) in this WP6 end-point series works. 

## Figures and Tables

**Figure 1 molecules-28-02767-f001:**
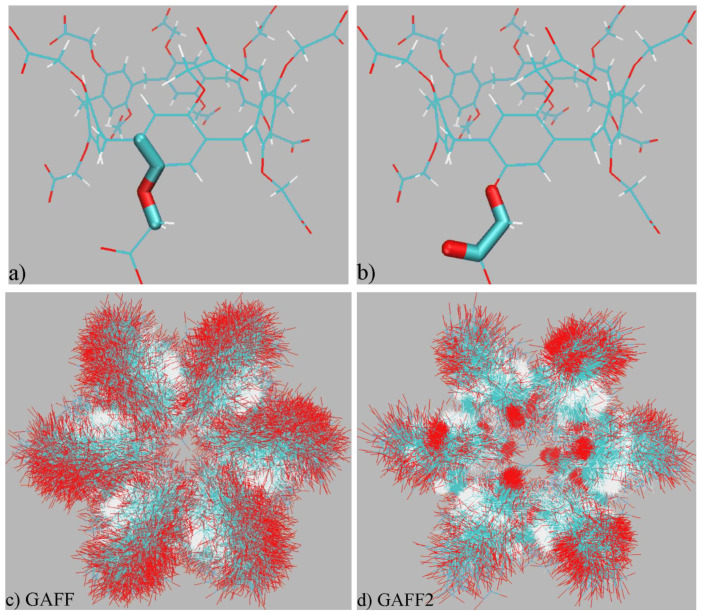
(**a**,**b**) Illustration of torsional terms that exhibit noticeable differences in the two GAFF versions. The GAFF and GAFF2 definitions of the (**a**) torsional term maintaining the planar structure share the same periodicity and phase, but the barrier height in GAFF2 is elevated by a factor of ~2 compared with GAFF. There is no explicit definition of the (**b**) torsion barrier in GAFF, but in GAFF2 this torsional interaction is defined by three Fourier-series terms with different periodicities (1, 2, and 3), phases, and barrier heights. The other five torsional potentials sharing the same definition under GAFF and GAFF2 are presented in Appendix A. (**c**,**d**) Superpositions of the host structure in 500-ns explicit solvent sampling under GAFF derivatives. The rotation of -CH_2_-COO^−^ tails exhibits some directional preference under GAFF2, while under GAFF a rather flexible dynamic behavior is observed.

**Figure 2 molecules-28-02767-f002:**
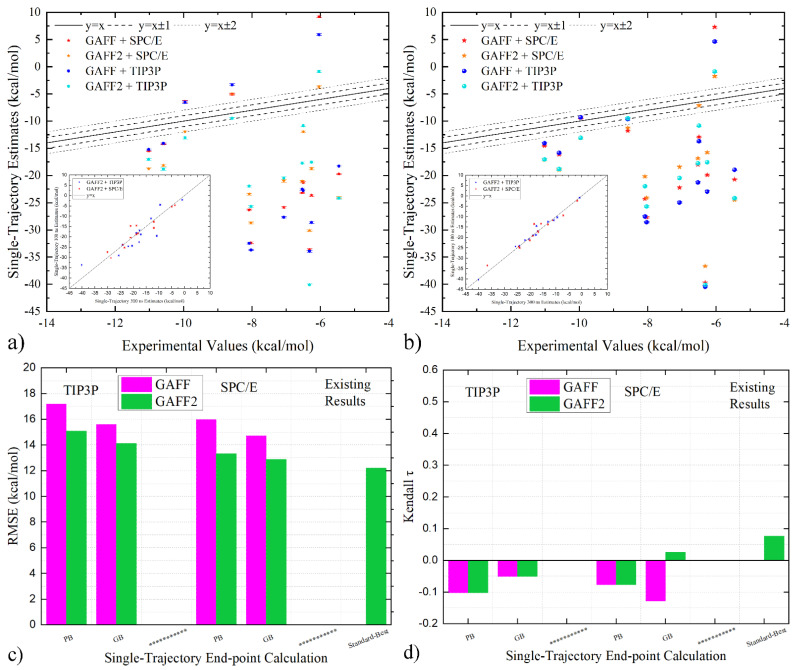
Correlation of single-trajectory end-point estimates and experimental reference under GAFF and GAFF2 with (**a**) PB and (**b**) GB implicit solvent treatment. The newly generated 300-ns trajectories are used in calculation, and the 100 ns GAFF2 results obtained from our previous work are compared in the inset of each subplot. (**c**,**d**) Comparison between quality metrics of single-trajectory end-point estimates under GAFF and GAFF2. It is clearly shown that in most cases the GAFF estimates are even less satisfactory than GAFF2. Therefore, all single-trajectory end-point estimates obtained from the straightforward calculation are not really useful in host–guest binding.

**Figure 3 molecules-28-02767-f003:**
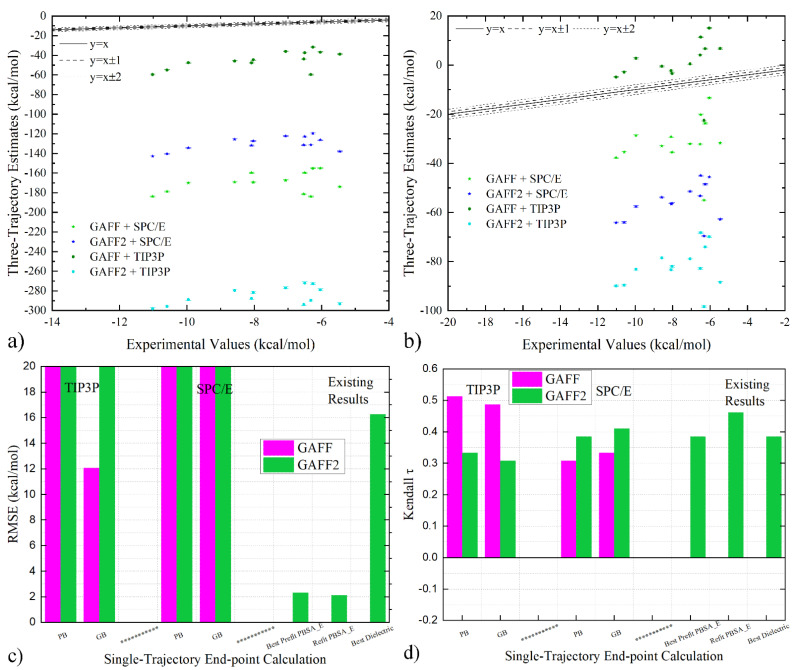
Correlation of three-trajectory estimates and experimental reference under GAFF and GAFF2 with (**a**) PB and (**b**) GB implicit solvent treatment. (**c**,**d**) Comparison between quality metrics of three-trajectory end-point estimates under GAFF and GAFF2. Three existing estimates obtained from modified end-point procedures reported in our previous works are also provided (specifically PBSA_E scoring functions pre-fitted with protein–ligand data and refitted with WP6-specific experimental data and the dielectric constant-variable regime).

**Figure 4 molecules-28-02767-f004:**
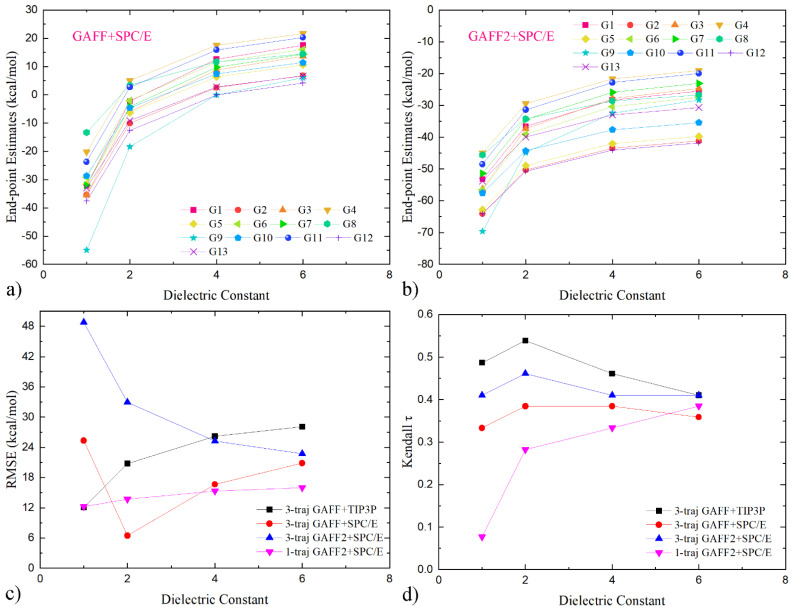
Dependence of three-trajectory MM/GBSA estimates on the dielectric constant for the modelling sets of (**a**) GAFF+SPC/E and (**b**) GAFF2+SPC/E. Clearly, the computed binding affinities of all host–guest pairs exhibit monotonic increasing behaviors with respect to the interior dielectric constant. This systematic increasing trend differs from the single-trajectory case, where system-dependent responding behaviors are observed. (**c**,**d**) Quality metrics for dielectric constant calculations with different modelling parameters. The single-trajectory results with the GAFF2+SPC/E parameter set are also shown for comparison. Under the three-trajectory regime, the response of RMSE seems to be parameter-set-dependent. The increase in the internal dielectric constant betters the reproduction of the absolute values of binding affinities under GAFF2+SPC/E but worsens the situation under GAFF+TIP3P, while under GAFF+SPC/E a non-monotonic trend is observed. Unlike the large variations in RMSE, minor perturbations are introduced to the ranking information. The ranking τ of the best-performing set GAFF+TIP3P in the three-trajectory realization can be bettered to ~0.54 with the adjustment of the interior dielectric constant.

**Figure 5 molecules-28-02767-f005:**
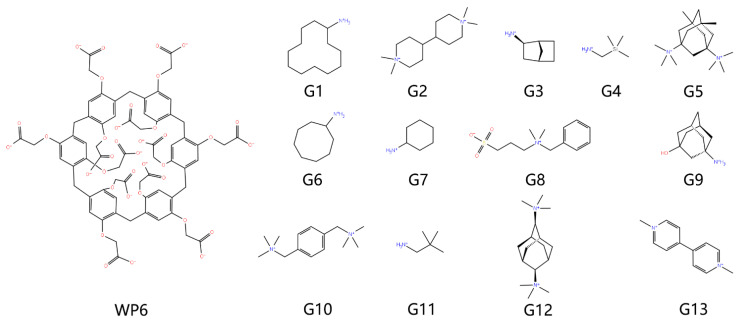
2D chemical structures of WP6 host–guest systems investigated in the current work. The macrocyclic host WP6 is forming one-to-one coordination complex with all 13 guests (from G1 to G13).

**Table 1 molecules-28-02767-t001:** Single-trajectory MM/PBSA estimates under GAFF and GAFF2. MSE, RMSE, τ, and PI serve as quality measurements.

Host	Guest	ΔG_exp_	GAFF				GAFF2			
			TIP3P	±	SPC/E	±	TIP3P	SD	SPC/E	±
WP6	G1	−6.53	−22.5	0.2	−23.2	0.1	–22.7	0.2	–21.1	0.2
	G2	–10.59	–14.1	0.2	–14.2	0.1	–18.1	0.2	–18.2	0.2
	G3	–8.03	–33.7	0.1	–32.4	0.2	–30.6	0.2	–28.7	0.2
	G4	–6.50	–22.8	0.2	–21.3	0.2	–18.2	0.2	–12.0	0.2
	G5	–5.46	–18.3	0.1	–19.7	0.1	–24.0	0.2	–24.1	0.2
	G6	–8.08	–32.5	0.2	–26.3	0.1	–25.7	0.2	–23.4	0.2
	G7	–7.07	–27.7	0.1	–25.9	0.2	–23.1	0.2	–21.0	0.2
	G8	–6.04	5.9	0.2	9.2	0.2	–2.4	0.2	–3.7	0.2
	G9	–6.32	–33.9	0.2	–33.6	0.1	–33.5	0.1	–30.1	0.1
	G10	–9.96	–6.6	0.2	–6.4	0.2	–11.8	0.2	–12.0	0.2
	G11	–6.26	–28.7	0.2	–23.7	0.2	–21.8	0.2	–18.8	0.2
	G12	–11.02	–15.2	0.1	–15.5	0.1	–19.0	0.2	–18.8	0.2
	G13	–8.58	–3.3	0.2	–5.1	0.1	–3.6	0.2	–5.0	0.2
RMSE			17.2		16.0		15.1		13.3	
MSE			11.8		10.6		11.9		10.5	
τ			–0.1		–0.1		–0.1		–0.1	
PI			–0.1		–0.1		–0.1		–0.1	

**Table 2 molecules-28-02767-t002:** Single-trajectory MM/GBSA estimates under GAFF and GAFF2. MSE, RMSE, τ, and PI serve as quality measurements.

Host	Guest	ΔG_exp_	GAFF				GAFF2			
			TIP3P	±	SPC/E	±	TIP3P	SD	SPC/E	±
WP6	G1	–6.53	–21.3	0.2	–18.0	0.1	–17.7	0.2	–16.8	0.2
	G2	–10.59	–15.8	0.2	–16.1	0.1	–18.8	0.2	–19.0	0.2
	G3	–8.03	–28.6	0.1	–27.7	0.2	–25.7	0.2	–24.1	0.2
	G4	–6.50	–13.7	0.2	–12.9	0.2	–10.8	0.2	–7.2	0.2
	G5	–5.46	–19.0	0.1	–20.7	0.1	–24.2	0.2	–24.5	0.2
	G6	–8.08	–27.5	0.1	–24.3	0.1	–22.0	0.2	–20.2	0.2
	G7	–7.07	–25.0	0.1	–22.2	0.2	–20.5	0.2	–18.4	0.2
	G8	–6.04	4.6	0.2	7.3	0.2	–0.9	0.2	–1.7	0.2
	G9	–6.32	–40.5	0.2	–39.7	0.1	–40.1	0.1	–36.7	0.1
	G10	–9.96	–9.3	0.2	–9.4	0.2	–13.1	0.2	–13.1	0.2
	G11	–6.26	–23.0	0.2	–19.9	0.2	–17.6	0.2	–15.8	0.2
	G12	–11.02	–14.1	0.1	–14.6	0.1	–17.0	0.2	–17.1	0.2
	G13	–8.58	–9.6	0.2	–11.8	0.1	–9.5	0.2	–11.3	0.2
RMSE			15.6		14.7		14.1		12.9	
MSE			10.9		10.0		10.6		9.6	
τ			–0.1		–0.1		–0.1		0.0	
PI			–0.1		0.0		–0.1		0.0	

**Table 3 molecules-28-02767-t003:** Three-trajectory MM/PBSA end-point estimates. MSE, RMSE, τ, and PI serve as quality measurements. The detailed components of free energy contributions are presented in the Appendix A.

Host	Guest	ΔG_exp_	GAFF				GAFF2			
			TIP3P	±	SPC/E	±	TIP3P	SD	SPC/E	±
WP6	G1	–6.53	–43.7	0.4	–181.3	0.5	–293.9	1.1	–131.3	0.5
	G2	–10.59	–54.9	0.4	–178.7	0.5	–295.8	1.1	–140.3	0.5
	G3	–8.03	–44.6	0.4	–169.4	0.5	–281.7	1.1	–127.3	0.5
	G4	–6.50	–37.4	0.4	–159.6	0.5	–272.0	1.1	–122.6	0.5
	G5	–5.46	–38.9	0.4	–174.0	0.5	–293.2	1.1	–137.9	0.5
	G6	–8.08	–47.7	0.4	–159.6	0.5	–287.8	1.1	–131.7	0.5
	G7	–7.07	–36.0	0.4	–167.4	0.5	–276.8	1.1	–122.3	0.6
	G8	–6.04	–36.8	0.4	–155.0	0.5	–279.0	1.1	–126.2	0.5
	G9	–6.32	–59.6	0.4	–183.9	0.5	–289.7	1.1	–131.2	0.5
	G10	–9.96	–47.5	0.4	–170.0	0.5	–288.9	1.1	–134.3	0.6
	G11	–6.26	–31.7	0.4	–155.2	0.5	–272.7	1.1	–119.5	0.6
	G12	–11.02	–59.6	0.4	–183.8	0.5	–298.2	1.1	–142.8	0.5
	G13	–8.58	–45.8	0.4	–169.1	0.5	–279.6	1.1	–125.5	0.5
RMSE			38.0		162.3		277.7		122.6	
MSE			37.2		162.0		277.6		122.5	
τ			0.5		0.3		0.3		0.4	
PI			0.6		0.4		0.4		0.5	

**Table 4 molecules-28-02767-t004:** Three-trajectory MM/GBSA end-point estimates. MSE, RMSE, τ, and PI serve as quality measurements. The detailed components of free energy contributions are presented in the Appendix A.

Host	Guest	ΔG_exp_	GAFF				GAFF2			
			TIP3P	±	SPC/E	±	TIP3P	SD	SPC/E	±
WP6	G1	–6.53	4.0	0.3	–32.2	0.2	–82.8	0.3	–53.2	0.3
	G2	–10.59	–2.9	0.2	–35.4	0.2	–89.6	0.3	–64.1	0.3
	G3	–8.03	–3.5	0.2	–35.4	0.2	–82.0	0.3	–56.3	0.3
	G4	–6.50	11.4	0.2	–20.2	0.2	–68.2	0.3	–45.0	0.3
	G5	–5.46	6.7	0.2	–31.7	0.2	–88.4	0.3	–62.8	0.2
	G6	–8.08	–2.3	0.2	–29.2	0.2	–83.3	0.3	–56.5	0.3
	G7	–7.07	0.5	0.2	–32.1	0.3	–78.9	0.3	–51.4	0.3
	G8	–6.04	15.1	0.3	–13.4	0.3	–69.9	0.3	–45.6	0.3
	G9	–6.32	–22.5	0.2	–55.0	0.2	–98.3	0.3	–69.6	0.2
	G10	–9.96	2.8	0.3	–28.7	0.3	–83.1	0.3	–57.6	0.3
	G11	–6.26	6.7	0.3	–23.7	0.3	–74.0	0.3	–48.5	0.3
	G12	–11.02	–4.9	0.2	–37.7	0.2	–89.9	0.3	–64.2	0.2
	G13	–8.58	–0.5	0.3	–33.0	0.2	–78.5	0.3	–53.8	0.2
RMSE			12.1		25.3		74.7		48.8	
MSE			–8.5		23.6		74.3		48.3	
τ			0.5		0.3		0.3		0.4	
PI			0.4		0.3		0.3		0.3	

## Data Availability

The atomic-charge files of all molecules and the docking-produced initial bound structures with the AD4 and Vina scoring functions are shared online at https://github.com/proszxppp/WP6-host-guest-binding, accessed on 22 February 2023. All free energy estimates obtained in this work are given in the main article and the Appendix A.

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
