# Peer review of "Comprehensive Evaluation of End-Point Free Energy Techniques in Carboxylated-Pillar[6]arene Host–Guest Binding: III. Force-Field Comparison, Three-Trajectory Realization and Further Dielectric Augmentation"

_molecules, 2023, doi:10.3390/molecules28062767_

Round 1

Reviewer 1 Report

The authors performed a series of computationally costly molecular dynamics simulation in order to assess the quality of a novel three-trajectory method to predict the binding energy of host/guest systems. Despite their effort, the final conclusion is that the gain in ranking (and not in terms of binding affinities) is only marginal. Yet, this might be due to the specific system chosen. Overall, the manuscript is well written, but the final result not so exciting. Nevertheless, it is worth of publication after major revision.

1. One of my main concern regards the use of HF to generate the RESP charges. It is known that higher level ab initio methods or even DFT will result in better description of the charges, and it will be doable also for the macrocycle. 

2. It is known that TIP3P does not properly reproduce water physical properties, and would be advisable to use a better method when considering the effect of solvent.

3. Fig 2 is hard to read; the background color should be changed and the contrast increased.

4. When comparing GAFF and GAFF2, the authors often describe the differences as 'insignificant'. This is too vague and should be quantified. In addition, did the authors perform a DFT (or other ab initio method) scan of the CH2COO dihedral to assess the quality of the GAFF2 parameterization?

5. It is not clear why the host solvent/ion interaction is not present for GAFF. Since there is free rotation of the CH2COO torsion, why this interaction is not seen? 

6. Fig 3 a,b is too dense and difficult to read. I suggest to move the previous results in SI, and the differences between the two used FF should be highlighted in the plots. I would be careful to sell GAFF2 as much better than GAFF, as the improvement is not high, for the standard method.

7. It seems that GAFF performs better with the three-trajectory method, while GAFF2 with the single one. Why is that? What is the advantage of the proposed method, in view of the more expensive computational cost? 

8. Is the novel proposed method transferable to different hosts or host/guest systems?

Minor remarks:

Abstract is too long.

The dimension of the PBC cell is not reported and the temperature details are missing. As it is now, the experiment is not reproducible by other groups, as too little information is given.

The difference between table 1 and 2 is not clear, and should be stated in the text.

Fig 4 has wrong label assignment (should be three-trajectory, not single-trajectory).

Author Response

Reviewer #1:

The authors performed a series of computationally costly molecular dynamics simulation in order to assess the quality of a novel three-trajectory method to predict the binding energy of host/guest systems. Despite their effort, the final conclusion is that the gain in ranking (and not in terms of binding affinities) is only marginal. Yet, this might be due to the specific system chosen. Overall, the manuscript is well written, but the final result not so exciting. Nevertheless, it is worth of publication after major revision.

  1. One of my main concern regards the use of HF to generate the RESP charges. It is known that higher level ab initio methods or even DFT will result in better description of the charges, and it will be doable also for the macrocycle.

Response: This is an interesting and important point. We should point out that this RESP parametrization has been benchmarked in the first paper of this WP6 end-point series, Journal of Computer-Aided Molecular Design 36 (10), 735-752. In that paper, we compute the molecular ESP generated by AM1-BCC charges, HF/6-31G*-targeted RESP charges, and a modern-level RESP set fitted at B3LYP/def2-TZVPP plus IEFPCM solvation, and then compare the charge-produced ESP data with those scanned at different ab initio levels (e.g., MN15/def2-TZVPP) and found that the HF/6-31G*-targeted RESP charges perform reasonably well and thus are applied in the following 2nd and 3rd (the current) papers. For more details, please read the first paper of this WP6 series.

  1. It is known that TIP3P does not properly reproduce water physical properties, and would be advisable to use a better method when considering the effect of solvent.

Response: TIP3P is widely applied in end-point simulations and thus is considered in our work. Note that we also tested another water model, SPC/E, in the current work.

  1. Fig 2 is hard to read; the background color should be changed and the contrast increased.

Response: We modified all subplots in Fig. 2 and S1 according to the reviewer’s suggestion, i.e., increasing the contrast and modifying the background color. Below is the comparison between the old Fig. 2a (left) and the adjusted version (right).

  1. When comparing GAFF and GAFF2, the authors often describe the differences as 'insignificant'. This is too vague and should be quantified. In addition, did the authors perform a DFT (or other ab initio method) scan of the CH2COO dihedral to assess the quality of the GAFF2 parameterization?

Response: The reviewer’s comment on scanning the CH2COO dihedral with DFT calculations is very reasonable and in an ongoing work we indeed do a similar job and even reparametrize the force field. However, concerning end-point free energy calculations, it would never be satisfactory to do a force-field refitting, as the end-point workflow itself thrives on efficiency (low cost) and the costly ab initio calculations in force-field refitting seems too demanding for practitioners to attempt in end-point calculations. Therefore, in the current WP6 end-point series, we focused on pre-fitted parameter sets (GAFF derivatives) in our benchmark calculations. Only in situations where the increase of computational costs is acceptable, e.g., using more rigorous and costlier enhanced sampling techniques to pursue high accuracy, the force-field refitting procedure would be attempted. As for some quantitative insights into the magnitude of parameter differences, we added some statistics of the average variations for the vdW and bond-stretching potentials (e.g., ~2% sigma change) to the last paragraph of section 3.1.

  1. It is not clear why the host solvent/ion interaction is not present for GAFF. Since there is free rotation of the CH2COO torsion, why this interaction is not seen?

Response: Host-ion interaction also exists under GAFF, but it does not incur significant conformational changes like the GAFF2 situation. This clarification has been added to the last paragraph of section 3.2 in the revision.

  1. Fig 3 a,b is too dense and difficult to read. I suggest to move the previous results in SI, and the differences between the two used FF should be highlighted in the plots. I would be careful to sell GAFF2 as much better than GAFF, as the improvement is not high, for the standard method.

Response: We improved Fig. 3a-b by removing the old 100 ns single-trajectory estimates, changing the coloring regime of the new 300 ns estimates, generating a correlation plot comparing the old 100 ns and new 300 ns GAFF2 estimates and merging it into the inset of Fig. 3a-b. As for the comment about selling GAFF2, it is related to the single-trajectory part of the paper. The overall tone for the single-trajectory realization is rather pessimistic, and in the paper and also all papers in this WP6 end-point series, we recommend not using the popular single-trajectory method regardless of the force field but are proposing to apply modifications. The GAFF2-better-than-GAFF statement in the single-trajectory case is simply an observation according to the numerical results. 

  1. It seems that GAFF performs better with the three-trajectory method, while GAFF2 with the single one. Why is that? What is the advantage of the proposed method, in view of the more expensive computational cost?

Response: The relative performance of GAFF and GAFF2 should be related to error cancellation. Personally, I think that end-point predictions under both force fields are not very satisfactory, but the numerical results for the current dataset simply suggest that one outperforms the other. Some clarifications about this issue have been added to the last paragraph of section 3.4. Note that we are currently working on the large-scaling computation of similar host-guest systems, which would provide a more stable and reliable view of the relative performance of these force fields. As for the advantage of the proposed method, relevant statements have been provided in the last paragraph of section 3.4 and that of section 3.5 in the current manuscript. Specifically, the three-trajectory and the further dielectric-constant-augmented methods provide ranking information comparable to costly alchemical method with polarizable force fields. 

  1. Is the novel proposed method transferable to different hosts or host/guest systems?

Response: Yes, the proposed three-trajectory dielectric-constant-variable regime is fully transferable to other host-guest systems. Currently, we are dealing with large-scaling calculations in other host-guest systems, which would be reported in a following paper.

Minor remarks:

Abstract is too long.

Response: The abstract has been shortened to 300 words that fit the journal’s requirement.

The dimension of the PBC cell is not reported and the temperature details are missing. As it is now, the experiment is not reproducible by other groups, as too little information is given.

Response: We added the solute-edge distance in solvation and the temperature for simulation to section 2 in the revised manuscript. Note that the parameter files (atomic charges) and initial configurations (Autodock-produced bound structures) have been shared online in the GitHub repository, and the other details (sampling length, water model and implicit solvent) have been provided in the main article. Our data sharing gesture already enables reproduction.

The difference between table 1 and 2 is not clear, and should be stated in the text.

Response: Clarifications added to the first paragraph of section 3.3.

Fig 4 has wrong label assignment (should be three-trajectory, not single-trajectory).

Response: This is a critical comment that requires careful reading. Thank you for your efforts and we have corrected this typo in revision.

Author Response

Submission ID: molecules-2276401

Responses to Referees’ comments

------------------------------------

COMMENTS TO AUTHOR:

Reviewer #2:

In the study, Liu et. al. performed MD simulations in gas phase and in solvent phase to evaluate the performance of different GAFF force-fields for host-guest binding energies. The study provides details on comparison of one-trajectory vs three-trajectory calculations and evaluates the performance of a combined regime of end-point free energy techniques. The manuscript is wellwritten and but needs to address following major issues before the publication:

Response: There is an error in this comment. We did not perform gas-phase sampling of any system. I do not know where the reviewer secures this information, but we should make it clear that we only performed explicit-water sampling of host, guest and host-guest complexes.

  1. The abstract in the present version is too long and detailed.

Response: The abstract has been shortened to 300 words that fit the journal’s requirement.

  1. The authors should outline in the introduction that there are other methods namely FEP

and TI for calculation of host-guest binding energies.

Response: We thank the reviewer for pointing out that there are many other methods usable for host-guest binding, i.e., FEP and TI. However, there is one conceptualization error in the reviewer’s statement. FEP and TI are just two post-processing methods, while the method referred by the reviewer should be called the alchemical method, which couples the physical end states with artificial transformation pathways. Actually, we have previously performed similar calculations in other host-guest complexes, see e.g., Journal of Computer-Aided Molecular Design 35, 117-129. In the current introduction, we have mentioned costlier free energy methods as rigorous free energy simulations in the fourth sentence of the introduction section. A note to add is that despite the rigor of this alchemical method, previous works by us (e.g., Carbohydrate Polymers 297, 120050 and Journal of chemical information and modeling 61 (12), 6107-6134) have already revealed that simple alchemical transformations cannot properly handle host-guest binding. The problems lie mostly in the multi-modal binding behavior of host-guest complexes, although other factors such as force-field parametrizations also play a role.

  1. Line 117 The authors state “Numerical results suggest that the three-trajectory realization serves as 116 a promising modification for ranking predictions”. The errors with respect to the experimental values are in the range of ~10-25 kca/mol (Table 1 and 2) for guests G1, G3-G9.

Response: I understand that the reviewer is worrying about the absolute values, but in the mentioned sentence we are stating that the three-trajectory realization is a promising method for ranking predictions. This statement is fully supported by solid numerical data presented in this paper. More detailed rebuttal about the absolute deviations is presented in the response to the next point.

  1. In Table 3, the calculated host-guest binding energies having a difference of more than 120 and 250 kcal/mol with SPC/E and TIP3P with GAFF2. Why do the authors want to publish results with these large variations?

Response: It should be noted that although accurate reproduction of absolute values is important, the end-point predictions are often used to predict ranking information, e.g., rescoring docking results and predicting the relative binding strength of different inhibitors to the same protein target (see references such as J. Chem. Inf. Model. 2021, 61, 6, 2844–2856). In relevant end-point papers for protein-protein binding (e.g., Briefings in Bioinformatics 23 (3), bbac127), the systematic errors could be even larger. Therefore, despite the differences in absolute values, the relative magnitudes of different host-guest pairs (i.e., the ranking predictions) are still meaningful. The Kendall tau is as large as 0.5 for the three-trajectory GAFF estimates. This value is comparable to costlier alchemical free energy calculations with polarizable force fields (see the first paper in this WP6 end-point series at Journal of Computer-Aided Molecular Design 36 (10), 735-752). Considering this encouraging fact, obviously the three-trajectory regime is usable.

  1. I don't find any strong conclusion in this work. The dihedral parameterization for acetate will not resolve their error of more than 20 kcal/mol. The authors should also enlist other contributing factors.

Response: The huge errors of absolute binding affinities are elucidated to be triggered by the implicit-solvent contributions, which has been investigated in Fig. S2-S4 in the current work. Specifically, the solute surface (Connolly surface here) is depicted for snapshots with different structural features (c.f., the last subplot of Fig. 2), and the change in this surface definition leads to significant variations of the implicit-solvent contributions. The reviewer’s comments on other contributing factors are reasonable, and in the following papers of this WP6 end-point series further detailed investigations of other influencing factors would be reported.

Minor edits:

1) The word “Comprehensive” from the title should be deleted, “comprehensive” might be misleading here considering the authors have only studied the two GAFF versions and the one- and three-trajectory calculations.

Response: As the reviewer mentioned, in this work we investigate the single- and three-trajectory and compare the two popular GAFF derivatives. A further point studied that is not mentioned in the reviewer’s comments is the combination of the three-trajectory realization and the dielectric-constant-variable regime, which is an advanced end-point method that is never explored in existing literatures. With many parameter combinations for the other computational setups (i.e., water model in explicit-solvent sampling, implicit-solvent model in free energy estimation), the three-trajectory and relevant methods are indeed comprehensively benchmarked. Further, the current work as the third paper is just reporting a subset of many aspects of the comprehensive investigation. Thus, we believe the ‘comprehensive’ statement in the title is reasonable.  

2) Line 600: Please replace the word “bettered” with “improved”

Response: The word ‘bettered’ has been modified to ‘improved’ in the revised manuscript. 

Reviewer 3 Report

The authors describe the host-guest binding, by comparing GAFF and GAFF2 force fields. They report the differences employing one vs three trajectories, using different types of water molecules, and considering the dielectric constant as variable.

The work is well described, and full of details. Overall, it is not easy to read, but for readers expert in the field is an interesting dissertation.

The topic is interesting, and the argument is new.

This is the first work of GAFF to GAFF2 comparison.

The references are appropriate

Author Response

Submission ID: molecules-2276401

Responses to Referees’ comments

------------------------------------

COMMENTS TO AUTHOR:

Reviewer #3:

The authors describe the host-guest binding, by comparing GAFF and GAFF2 force fields. They report the differences employing one vs three trajectories, using different types of water molecules, and considering the dielectric constant as variable.

The work is well described, and full of details. Overall, it is not easy to read, but for readers expert in the field is an interesting dissertation.

The topic is interesting, and the argument is new.

This is the first work of GAFF to GAFF2 comparison.

The references are appropriate

Response: We appreciate experts’ opinions on our manuscript. Thank you for your review.

Round 2

Reviewer 1 Report

The authors replied to all my comments in a satisfactory way and added more information in the manuscript.

They have green light for publication from my side.

Reviewer 2 Report

publish after accepting these changes.